# Secondary Traumatic Stress, Religious Coping, and Medical Mistrust among African American Clergy and Religious Leaders

**Laura Roggenbaum** [1], **David C. Wang** [2,*], **Laura Dryjanska** [1], **Erica Holmes** [3], **Blaire A. Lewis** [4] **and Eric M. Brown** [5]

1 Rosemead School of Psychology, Biola University, La Mirada, CA 90639, USA; laura.roggenbaum@biola.edu (L.R.); laura.dryjanska@biola.edu (L.D.)
2 Fuller Theological Seminary, Pasadena, CA 91182, USA
3 Champion Counseling Center at Faithful Central Bible Church, Los Angeles, CA 90047, USA; elholmes@faithfamily.org
4 Independent Researcher, Chicago, IL 60603, USA; blewis@drblaire.net
5 School of Medicine, Boston University, Boston, MA 02215, USA; ebrown1@bu.edu
* Correspondence: davidcwang@fuller.edu

**Abstract:** Previous research has investigated the prevalence and impact of secondary traumatic stress (STS) among those working as helping professionals. However, limited studies have provided clear and coherent information about STS among clergy, pastors, and other religious leaders, despite their status as helping professionals who are implicated in times of crisis. STS is particularly salient to African American religious leaders due to cultural factors that position African American churches as trusted institutions linking local communities of color with various social services. Results from a sample of African American religious leaders confirmed the prevalence of STS along with other mental health challenges. Moreover, STS was associated with negative interactions within the church. Finally, negative religious coping and medical mistrust significantly moderated the relationship between adverse childhood experiences and PTSD. These findings bear significant implications, emphasizing the need for greater collaboration and trust-building between mental health professionals and clergy.

**Keywords:** African American clergy; secondary traumatic stress; adverse childhood experiences; post-traumatic stress disorder; medical mistrust





## 1. Introduction

As churches play a fundamental role in offering care for their congregants, trauma researchers have begun to study the impact of traumatic experiences among religious leaders (Davis and Johnson 2020). Secondary traumatic stress (STS) is defined as the development of symptoms that often accompany trauma exposure through indirect means, such as hearing about the trauma (Figley 1995). The symptoms experienced in STS are similar to the symptoms of post-traumatic stress disorder (PTSD; Figley 1995). While STS is typically studied among populations of helping professionals, such as social workers and law enforcement officers, who are frequently exposed to the trauma of their clientele, religious leaders also play a critical supportive role in their spiritual communities as well as in general society, potentially putting them at risk for STS as well. Moreover, African American religious leaders play a unique role in offering trauma care, as historical and current racism has exacerbated wariness among African Americans to seek therapeutic support from traditional mental health resources (Hays 2015).

## 2. STS in Religious Leaders

The literature exploring the negative impact of doing trauma work has highlighted the prevalence and salience of STS. In a survey of social workers, Choi (2011) found the

existence of one or more STS symptoms in most of the respondents, with one-third of the social workers indicating moderate to severe symptoms. Overall, mild levels of PTSD were reported among the respondents; however, 65% of the participants testified to the occurrence of at least one symptom of STS. Over half of the participants informed the researchers of intrusive symptoms, and approximately 21% reported intrusion, avoidance, and arousal (Choi 2011).

Similarly, a study conducted by Bride (2007) on a separate sample of social workers found that 70.2% experienced at least one symptom of STS in the prior week. Of these participants, 15.2% met the criteria for PTSD and 55% met the criteria for one of the main symptom clusters. Intrusive thoughts, avoidance of clients, and emotional numbing were the most prominently reported symptoms. In this pool of participants, 40.5% reported thinking about their work with clients without meaning to do so. Though 45% of social workers did not experience STS symptoms, over half the sample met at least one criterion for PTSD, and 15.2% met all the central diagnostic criteria. Finally, 86.7% of the respondents indicated feeling fear, horror, and helplessness in response to hearing about the trauma of their clients. Overall, research findings have suggested the high susceptibility of those who work closely with traumatized individuals to STS. However, documentation of the prevalence of STS in African American religious leaders is lacking.

Church congregants often seek out religious leaders for support in the aftermath of traumatic experiences (Wang et al. 2014). Clergy members are typically in long-term relationships with individuals and their families in which they assist in providing specialized care not only immediately following traumatic experiences but also for those facing mental illnesses such as anxiety, depression, and PTSD. A 2003 study reported that clergy were called upon for such frontline work in greater amounts than even psychiatrists or medical doctors; notably, approximately 25% of those seeking treatment for mental disorders did so from the clergy (Wang et al. 2003). Furthermore, clergy acted in noteworthy supportive roles following the 9/11 terrorist attacks, which led to the development of significant levels of compassion fatigue for some of the clergy (Flannelly et al. 2005; Taylor et al. 2006).

Another study that examined 99 Black pastors discovered that 40% of the pastors encountered severely mentally ill congregant members. Additionally, two-thirds of the pastors reported counseling congregants with suicidal proclivities, with ten percent of these involvements being linked to a crisis (Payne 2014). Moreover, religious leaders are often called upon as a means for care and support for those who have faced or are currently facing the trauma of intimate partner violence and abuse as well as sexual abuse (Davis and Johnson 2020; Rudolfsson and Tidefors 2009; St. Vil et al. 2016). In a qualitative study of thirteen African American clergy members, each one described having been contacted for guidance on this subject during their tenure (Davis and Johnson 2020). Black women frequently mention religion as an important basis when looking for assistance to deal with intimate partner violence and abuse. In addition to their belief in God and their religion, many Black women shared in a semi-structured interview that clergy or other members of the church helped them bear or overcome the violence experienced by their abusers (St. Vil et al. 2016).

The scores of religious leaders on common measures of STS symptomology have resembled the scores of mental health professionals (Hendron et al. 2011; Holaday et al. 2001). In a qualitative interview study of Irish clergy who work with trauma, half of the clergy were experiencing a high number of symptoms of STS, and half were experiencing a low number of symptoms. The clergy described the emotional, behavioral, physical, familial, and spiritual impacts of STS. They indicated feeling emotionally overwhelmed, guilty, sad, distanced, and detached, as well as having trouble sleeping and nightmares. Furthermore, some suggested tertiary traumatization as they described the strain put on their spouses as they listened to upsetting accounts. Though none signified that their faith in God had been negatively tainted, some stated that their faith in organized religion was damaged (Hendron et al. 2011). A study of 1458 Roman Catholic ministers found that high levels of burnout existed within the sample. Over a third of the Catholic priests stated that

they felt used up after a day of ministry and felt that parishioners blame them for some of their problems. Additionally, over a quarter stated that they are working too hard in their ministry, working with people all day puts a burden on them, they are less patient with parishioners than they used to be, and they find it difficult to attend to parishioners (Francis et al. 2004).

The challenges that Black church leaders face related to burnout and STS are compounded by the need to shepherd congregants whose lives are deeply affected by living within racially unjust systems. In addition to the trauma faced by their congregants, many pastors are also processing their own experiences of racial trauma. Not only are they caring for their communities, but they also must find the time and resources to care for themselves. Baum (2014) found that when professionals faced double exposure due to the shared traumatic reality of the times, the main qualities of double exposure were significantly related to primary traumatization, secondary traumatization, and general symptomatology of distress. The key features of double exposure include intrusive anxiety, lapses in empathy, and changes in place and time of work (Baum 2014). Additionally, those pastors serving in a "traumatic church", which is categorized by elevated stress and conflict, were expected to have high emotional exhaustion that carried on even after they left that congregation (Doolittle 2010).

Though it is evident that support is vital for helping professionals to aid them in coping with the struggles associated with providing care to traumatized individuals, clergy often differ from mental health professionals in their access to support and consultation services. Mental health professionals often have supervision and peer support built into the model of their practice and training. They are trained to recognize and care for the personal stress that comes with the emotional burden of caring. However, clergy often lack spaces to discuss the problems they encounter as well as their reactions to these problems (Holaday et al. 2001). Furthermore, dual relationships cause issues for clergy in a way that mental health professionals do not often have to face. The numerous roles that pastors play in their communities sometimes constrain their ability to form intimate friendships within their church communities (Holaday et al. 2001). Additionally, religious leaders typically receive limited training in basic counseling skills and providing trauma-informed care. Many have conveyed the wish that they had structured training in areas regarding mental health care. Moreover, many religious leaders have communicated that they remain interested in receiving further education about these subjects (Holaday et al. 2001).

## 3. ACEs, Trauma, and STS in African American Communities

In conjunction with the Centers for Disease Control, Felitti et al. (1998) created the ACEs Questionnaire to explore the impact of trauma and other stressful experiences in childhood. Since the creation of the questionnaire, ACEs have become a major construct in mental and physical health research, which has led to countless studies showing the lingering negative physical and mental health outcomes of such adverse experiences (Mosley-Johnson et al. 2021). In comparison with White individuals, those identifying as Latinx, Black, and American Indian are more likely to experience over four ACEs (Assini-Meytin et al. 2021). Conversely, benevolent childhood experiences (BCEs) have recently emerged as a construct that counters some of the deleterious effects of ACEs (Bethell et al. 2019; Narayan et al. 2021).

Although White people experience depression and other mood disorders more frequently, research suggests that African Americans may experience them with increased severity and persistence (Pérez Benítez et al. 2014; Williams 2008). Historical institutional slavery and continuing expressions of racism continue to negatively impact the quality of life of African Americans. Depression and trauma often go hand in hand, and these societal issues compound the stress of African Americans. The stress and the consequences of trauma that many African Americans experience daily frequently aggravate these symptoms even further (Streets 2015; Williams and Mohammed 2009; Woods-Giscombe 2010). Many individuals within the Black community experience depression as a symptom that is

secondary to prior trauma exposure. Thus, their situation does not always fit deftly into the DSM-V criteria for depression or the diagnosis of PTSD but rather bears some facets of both conditions (Streets 2015).

PTSD and other trauma spectrum disorders can be extremely incapacitating and trouble African Americans at disproportionately higher rates than the rest of the population (Alegría et al. 2013; Williams et al. 2014). The lifetime prevalence of PTSD in African Americans was measured as 8.7 percent. Non-Hispanic European Americans, Hispanics, and Asians experience a lifetime prevalence of PTSD at 7.4 percent, 7.1 percent, and 4.0 percent, respectively (Roberts et al. 2011). Though the number of traumatic events documented by African Americans is lower than those of other races and ethnicities, the degree of the traumatic events is often more severe and violent. Common occurrences include assaultive violence, such as domestic violence, rape, homicide, physical attacks, and kidnappings, as well as exposure to child abuse. On the contrary, European Americans are more likely to report a traumatic experience such as learning of the unexpected death of a loved one (Asnaani and Hall-Clark 2017; Pérez Benítez et al. 2014). Moreover, racism, discrimination, and stigmatization that frequently occur in addition to other traumatic events may result in a composite reaction to the trauma, influencing the symptom severity. Studies have reported that increased perceived discrimination makes one more apt to a PTSD diagnosis or greater severity of PTSD. Furthermore, increased rates of hypervigilance and emotions such as guilt, shame, and low self-worth, which are also associated with PTSD, have been evident based on racial discrimination (Alegría et al. 2013; Asnaani and Hall-Clark 2017; Williams et al. 2014).

## 4. Barriers and Mistrust toward Healthcare

Research has continually identified a gap in the diagnosis and treatment of PTSD among African Americans. A large degree of racial discrepancies in PTSD prevalence and the rate of seeking treatment for PTSD are the result of the hesitation among many minority groups to utilize professional psychological services. African Americans are one-third to one-half as likely to consult a mental health provider, almost twice as likely to prematurely discontinue therapy, and three times more likely not to begin therapy in comparison with European Americans (Fiscella et al. 2002; Spoont et al. 2015; Williams et al. 2014). Research findings have indicated that African Americans with mental illnesses make healthcare appointments later in the progression of their illnesses and arrive more disabled than European Americans (Williams and Earl 2007). African Americans experience lower rates of recovery from PTSD. Several factors contribute to this disparity, including less access to financial resources for mental health care as well as over-representation in lower socioeconomic and underprivileged communities (Hays 2015; Pérez Benítez et al. 2014). However, even after adjusting for predisposing factors such as insurance availability, Black patients are significantly less likely to have had an appointment with a mental health provider (Fiscella et al. 2002).

Though African Americans account for thirteen percent of the population, they only comprise four percent of psychologists (Lin et al. 2018). They endorse greater levels of mental health stigma, more negative attitudes regarding mental health treatment, and increased fears of being discriminated against by mental health providers. Furthermore, the racial incompetency of some therapists serves as a barrier to treatment for many. They may also feel differently towards Caucasian clinicians due to cultural mistrust (Williams 2008; Williams et al. 2014). Mental health professionals often do not understand the impact of racism on psychopathology because of a societal tendency to deny or rationalize the presence of racism (Williams et al. 2014).

Past and recent experiences of discrimination in healthcare settings have led African American patients to demonstrate lower trust in the healthcare system than European Americans. Experiences of discrimination largely account for the existence of this mistrust. Even after studies have adjusted for healthcare disparities, there is still a large degree of association between the African American race and distrust in the healthcare system

(Armstrong et al. 2013; Cuevas and O'Brien 2019). In many cases, greater perceived racism and greater mistrust in the healthcare system leads to poorer health outcomes for African Americans. Mistrust of physicians has been seen to influence African American patients in such a way that they are less forthcoming with doctors about health information, more submissive to doctors, and less likely to follow treatment protocol. Overall, mistrust of the healthcare system results in decreased patient satisfaction with care and the utilization of services (Hammond et al. 2010a, 2010b; Peek et al. 2010).

Cuevas et al. (2016) conducted a qualitative study recording the first-hand healthcare experiences of African Americans. Participants described experiencing discrimination from both the office staff and doctors. For example, African American women stated that they had to continually declare their interests to obtain reasonable treatment. They also spoke of incidents in which White doctors did not try to take the time to listen to their concerns or ensure they comprehended their treatment (Cuevas et al. 2016). This sentiment is corroborated by a study that found that patients that were the same race as their physicians had visits that were an average of ten percent longer than those of differing races (Cooper et al. 2003). Moreover, African American men relayed experiences in which mistrust was connected to perceived discrimination and poor communication. Overall, the results of this study suggested that the healthcare system is structured in such a way that African Americans feel depersonalized and objectified. Healthcare visits feel rushed and disrespectful and lack advocacy from the doctor. Thus, a cycle of mistrust is created in which African American patients do not trust their doctors and believe that their doctors do not trust them (Cuevas et al. 2016).

African American patients may also have developed a mistrust of healthcare institutions in response to physician bias and discrimination as well as cultural discordance, which in turn impact the physician's behaviors. For example, physicians may be less likely to share medical information and more likely to be authoritarian toward African American patients. A lack of cultural competence among physicians is often displayed both verbally and nonverbally (Kennedy et al. 2007; Peek et al. 2010). In a study of race-concordant and race-discordant medical encounters, healthcare visits in which the doctor and patient were of the same race received higher coder ratings of positive affect-reflections of voice tone qualities, which is suggestive of the emotional framework of the appointment. These visits were further depicted as having greater scores of patient satisfaction and positive judgments of physicians' use of shared decision making. Patients and physicians that are members of the same race or ethnic groups are more likely to have similar beliefs, values, and experiences in society, which results in increased security with each other (Cooper et al. 2003; Kennedy et al. 2007).

It is vital not to interpret findings regarding mistrust of the healthcare system as a problematic characteristic of a patient. Often, factors that relate to mistrust, such as identification with one's racial group, can be imperative to one's mental health and well-being. Medical mistrust is not simply an attitudinal obstacle. Instead, it is a barrier that is prompted by proximal, first-hand experiences (Cuevas and O'Brien 2019; Powell et al. 2019). Mistrust can very well be a reasonable and adaptive reaction to historic and ongoing discriminatory experiences. Though there are negative health outcomes linked to mistrust of healthcare systems, it can also safeguard a person by circumventing some race-based stressors (Cuevas and O'Brien 2019).

## 5. The African American Church

Most African Americans are either past or current churchgoers, with approximately 85 percent identifying as fairly or very religious and 80 percent praying daily (Taylor et al. 2004). A sizable margin of 70 to 80 percent of African Americans denoted on one national survey that religion is an important component of their lives. Out of all racial and ethnic groups in the United States, African Americans have the highest documented rate of church attendance (Adofoli and Ullman 2014; Taylor et al. 2004; Williams 2008; Williams et al. 2014). The Black church in the U.S. has always served as a spiritual, psychological, and social

bulwark against anti-Black racism and a prophetic witness to the humanity of Black person-hood. The Black church has been a refuge throughout slavery to Freedom, reconstruction, Jim Crow, civil rights, and the discrimination of the current times (Streets 2015; Williams 2008). During slavery, the church, by law, was the sole forum where African Americans could assemble in groups and have a sense of freedom. The teachings, songs, and social support found in the church brought hope and confirmed humanity. It has been a voice advocating for social change and equality. The Black church has also set an example of the way that living a life of faith, values, and ethics can assist in handling injustice and adversity (Plunkett 2014; Williams 2008). Moreover, the Black church has continuously afforded tangible support services to people in their community by offering food, shelter, financial assistance, and direction (Streets 2015; Wingfield 1988).

African Americans often use faith-based resources for their mental health care over other clinical resources (Adofoli and Ullman 2014; Hankerson and Weissman 2012; Hays 2015). Some have called the Black church "a spiritual hospital". The Black church has undertaken the position of a therapist to tackle numerous emotional, psychiatric, and psychological mental health issues among its congregants. Seeking support and healing from the church is often more appealing than working with a mental health professional. Whereas the language of psychology may be foreign, the spiritual language of healing and transformation found in the church has deep cultural resonance (Plunkett 2014; Williams 2008). Faith-based programs offered through churches to promote mental and physical healthcare are culturally fitted to highlight Black culture and spirituality (Hankerson and Weissman 2012). Furthermore, the pastor of the church is someone that the congregants already trust, contrasting with a therapist with whom one must build up a relationship and trust. They are also familiar with the life circumstances and resources of the congregants and are thus better able to anticipate and respond to secondary issues. Moreover, both therapy and the church meet weekly; however, church is free (Payne and Hays 2016; Williams 2008). At large, many African American congregants see formal counseling as unwarranted because they often set their hope for healing in the promises of the Bible and prayer. Professional counseling is thus sometimes seen as a lack of faith (Plunkett 2014). Further, Payne (2009) found that African American pastors were more likely to concur that the etiology of depression was a lack of trust in God rather than a biological disorder. African American religious services have continued to operate as imperative social events, tending to shield individuals from feelings of estrangement and helplessness. Attendance at religious services has been negatively correlated with psychological distress in this community (Jarvis et al. 2005). Generally, the literature has shown that resources from the church are largely effective in symptom reduction; however, more research is needed in this area (Hankerson and Weissman 2012).

Two of the highest contributors to the disposition toward seeking mental health care among African American congregants are the beliefs of the religious leader and church doctrine. Clinicians are often unable to sufficiently address the cultural differences at play such as their religious and spiritual beliefs and the ideologies that impact mental health treatment (Hays 2015). African Americans who seek professional therapeutic help eventually turn to a pastor also. However, those who seek the pastor first turn to few additional resources afterward (Neighbors et al. 1998).

The number and types of crises and disruptions that antagonize the congregants of African American churches result in pastors accepting various leadership roles for which they are not always equipped. The expectation to be there for the congregants, serve as an outstanding pastor, and always display integrity leads to added stress on pastors (Streets 2015).

Some have raised the concern that church-based mental health care, provided by non-professionals, could be harmful to the congregants. A sizeable number of clergy are unqualified to handle mental health crises. Many pastors report feeling inadequate to deliver such care. Moreover, some hold beliefs that discourage seeking help from mental

health professionals. Further education and training are needed for religious leaders in mental health services (Asamoah et al. 2014; Hall and Gjesfjeld 2013).

The African American church's role in guidance, support, and communication places it in a unique and pivotal position for healthcare promotion efforts. The unique situation of the church as well as its concern for the well-being of its congregants can lead to partnerships that may reduce health disparities (Goldmon and Roberson 2004). Prior research has indicated that faith-based interventions have been impactful for smoking cessation, reductions in cardiovascular disease, and increasing the consumption of healthy foods among African Americans (Whitt-Glover et al. 2008). Moreover, a general readiness to change has been observed following these church-based interventions (DeHaven et al. 2004). Of note, barriers remain in the implementation of church-based health programs within the African American church. For example, historically, due to the separation of church and state, religious organizations were limited in acquiring the required funding to facilitate such programming (Lasater et al. 1997).

## 6. Religious Coping

Religiosity and spirituality have been predictors of mental health outcomes. The constructs have been associated with the increased ability to cope with stress and decreased burnout in the workplace (Graham et al. 2001; Perera et al. 2018). Moreover, Figley (1995) recognized the vitality of attending to the well-being of those at risk for STS and noted the helpful effects of spirituality. Feeling "spiritually dry", on the other hand, surfaced as a principal predictor of emotional exhaustion in a study on burnout among pastors (Chandler 2009). However, there have also been immense challenges evoked as some have attempted to reconcile trauma with one's experience of the divine. Trauma has either stimulated or demolished the spirituality in some people (Figley 1995). A study on faith leaders working with survivors of trauma in Columbia evaluated the spiritual struggles of the leaders. Those struggling spiritually had increased levels of STS (Currier et al. 2019).

Spirituality is used among African Americans to address mental-health-related concerns. Adofoli and Ullman (2014) found that positive religious coping was associated with less alcohol consumption in African American females who had been sexually assaulted. However, in a longitudinal study of African Americans who had experienced trauma, religious coping did not reduce PTSD symptoms, which led researchers to wonder if religious coping could exacerbate PTSD symptoms (Bryant-Davis et al. 2011, 2015). For some reason, religious coping in this sample exacerbated distress, which led researchers to hypothesize potential harmful uses of religious coping. Negative religious coping is characterized by the belief that God is punishing one for their actions, that one must wait passively on God for change, that one's faith is not strong enough, and that prayer is an avoidance tool. Moreover, some religious teachings blame the victim, which increases distress (Bryant-Davis et al. 2011). Subsequent research has also discovered associations between negative religious coping and PTSD, depression, and self-esteem in both predominately Caucasian and African American populations (Park et al. 2018; Walker et al. 2021; Wood et al. 2021). Researchers have hypothesized that when individuals fail to make meaning, particularly with religion, after experiencing trauma, there are adverse mental health outcomes. Additionally, trauma may threaten religious coping by triggering maladaptive religious coping systems (Park et al. 2018; Walker et al. 2021; Wood et al. 2021). On the other hand, there are mixed findings regarding research on positive religious coping. While some studies have indicated relationships between positive religious coping and mental well-being (Park et al. 2018), most studies have found no relationship between the two variables (Walker et al. 2021; Wood et al. 2021).

## 7. The Present Study

Given the pervasiveness of trauma, churches will encounter survivors of trauma and/or ACEs at some point—either among the leadership body and/or among the congregation and community at large. Since past and present experiences of discrimination

in healthcare settings have led African American patients to mistrust the healthcare system, African Americans often use faith-based resources for their mental health care over other clinical resources (Adofoli and Ullman 2014; Hankerson and Weissman 2012; Hays 2015). The African American church carries a unique role in the lives of its congregants in supporting health and wholeness. A review of the literature further reveals that religious leaders are exposed to traumatic material at rates comparable to other helping professionals recognized as at risk for STS (Hendron et al. 2011; Holaday et al. 2001). Yet, there are minimal studies specifically confronting the experience of STS in the clergy. Without this material, religious leaders may not be adequately conscious of, equipped for, or handling STS (Hendron et al. 2011).

The unique context of the African American church and its engagement with the holistic well-being of its congregants can lead to partnerships that may reduce health disparities (Goldmon and Roberson 2004). For congregation members, factors such as religiosity and attitudes toward healthcare can directly relate to their willingness to engage in formal healthcare settings. Therefore, it could prove vital to study the mechanisms that lead to mistrust and endeavor to address them through partnerships between congregations and mental health professionals. Accordingly, the present study aims to examine (a) the prevalence of ACEs, BCEs, STS, depression, anxiety, and PTSD; (b) the relationship between STS and church interactions; and (c) the moderating roles of both negative religious coping and medical mistrust on the relationship between ACEs and PTSD.

## 8. Method

### 8.1. Participants

One hundred and twenty-four males and females from a network of predominantly African American churches in Texas were recruited to participate in this study. Responses with greater than 30% missing data were excluded. Therefore, data from 102 participants were included in the study.

Seventy participants (68.6%) identified as female, and thirty-one participants (30.4%) identified as male. Participants were primarily between the ages of 45 and 74 years old (72.6%). Sixty-eight participants were married (66.7%), thirty-three were not married (32.4%), and one participant preferred not to say. In total, 75 participants reported holding some sort of leadership position in the church, with 9 identifying as a head pastor, 12 identifying as an associate pastor, and 4 identifying as a worship pastor. Other common leadership positions included elders (6 participants), deacon/deaconesses (5 participants), children/youth ministry leaders (7 participants), small group leaders (16 participants), counseling/prayer team members (7 participants), and administrators (3 participants). Additionally, five participants identified supportive volunteer positions including greeting, technological assistance, finance, and maintenance.

### 8.2. Procedure

Before recruiting participants, ethical approval for the study was obtained from the Institutional Review Board of a private university located in the western United States. Participants were recruited through a network of churches in Texas taking part in a faith-based, trauma-informed mental health initiative led by a non-profit organization with decades of experience addressing social inequalities among religious congregations. A group of African American church leaders met for a retreat in Fort Worth, Texas, where they completed the initial version of the survey utilized in this study. Following their feedback, the survey was dispersed via an online survey to their respective congregation leaders via email. Before completing the measures, participants agreed to informed consent and provided demographic information. Upon the completion of the survey, the participants were provided with the contact information of the researcher as an option to debrief the experiment as well as resources on trauma. Participants were given the option to enter their email addresses to receive compensation in the form of an electronic gift card. The survey took approximately 20 min to complete.

## 9. Measures

### 9.1. ACEs

A 10-item scale developed by Felitti et al. (1998) called the Adverse Childhood Experiences Scale (ACEs) was used to investigate the occurrence of acutely stressful and traumatic events that happened before the age of 18, to which participants respond "yes" (1) or "no" (0). The sum score was calculated, and scores ranged from 0 to 10. Example items include questions related to physical abuse, sexual abuse, and neglect. Internal consistency was 0.78 in the current study.

### 9.2. BCEs

The 10-item Benevolent Childhood Experiences Scale (BCEs; Narayan et al. 2018) was used to measure the positive childhood experiences of participants before the age of 18. Items include questions about safety and security, quality of life, and interpersonal connection throughout childhood and adolescence. Participants respond "yes" (1) or "no" (0), and scores are summed, creating a range of 0 to 10.

### 9.3. Anxiety Symptoms

The Generalized Anxiety Disorder Scale (GAD-7; Spitzer et al. 2006) was used to assess the presence of anxiety symptoms in respondents. The GAD-7 consists of seven items about the core anxiety symptoms and is evaluated on a 4-point Likert-type scale regarding the consistency of the symptoms. The scale ranges from *not at all* (0) to *nearly every day* (3). Scores range from 0 to 21, with higher scores indicating higher anxiety. The measure displayed good reliability, as demonstrated by an internal consistency score of 0.86. Clinical severity scores are as follows: 1–5, mild anxiety; 6–10, moderate anxiety; and 15–21, severe anxiety.

### 9.4. Depression Symptoms

Depression was assessed using the PHQ-9 (Kroenke and Spitzer 2002). Respondents are asked how often they have been bothered by nine depressive symptoms over the last two weeks on a 4-point Likert-type scale ranging from *not at all* (0) to *nearly every day* (3). Possible scores range from 0 to 27. Higher scores indicate higher depression. The PHQ-9 has demonstrated superior criterion validity in comparison to other self-report measures related to depression, which was consistent with the Cronbach's alpha score of 0.91 in the current sample. Consistent with cut-off scores for clinical use, the following score categories were used: 1–4, minimal depression; 5–9, mild depression; 10–14, moderate depression; 15–19, moderately severe depression; and 20–27, severe depression.

### 9.5. PTSD Symptoms

PTSD was assessed using the Primary Care PTSD Screen for *DSM-V* (PC-PTSD-5; Prins et al. 2016). The measure consists of five items designed to assess how trauma has affected the participant over the past month. Participants answer the questions by responding *yes* (1) or *no* (0). Scores range from 0 to 5. The reliability score for this measure in the current study was 0.86.

### 9.6. STS Symptoms

STS in the original 23 church leaders was measured using the Secondary Traumatic Stress subscale of the Professional Quality of Life Scale (ProQOL; Stamm 1995). The 10-item subscale assesses participants' responses to secondary exposure to traumatically stressful events, particularly at work. The current study replaced the word "helper" with "pastor/church leader". Some examples of items include "I am preoccupied with more than one person I help" and "As a result of my work as a pastor/church leader, I have intrusive, frightening thoughts". Each item is rated on a Likert scale of 1 (never) to 5 (very often). Scores range from 5 to 50. Reliability in the present study was 0.78. As suggested by Stamm, the following score categories were used: 5–22, low; 23–41, moderate; and 42–50, high.

### 9.7. Negative Religious Coping

The Negative Religious Coping subscale of the Brief RCOPE (Pargament et al. 2011) was administered to assess negative religious coping. There are seven items rated on a 4-point Likert-type scale ranging from *not at all* (0) to *a great deal* (3). Participants were prompted to respond to a variety of statements indicating the extent to which they have engaged in each in response to a critical life event. Statements included: "Felt punished by God for my lack of devotion" and "Decided the devil made this happen". Scores on the seven items were summed, creating an overall negative religious coping score ranging from 0 to 21. The subscales of the Brief RCOPE have shown good psychometric properties in a variety of contexts. In the present study, the negative coping subscale had an alpha of 0.85.

### 9.8. Negative Religious Interaction

Negative Religious Interaction was measured based on the two-item subscale from a measure of religious support developed by Fetzer Institute (1999). The scale acknowledges that interactions within the church are not always affirming and asks, "How often do the people in your congregation make too many demands on you?" and "How often are the people in your congregation critical of you and the things you do?". Responses are made using a 4-point Likert scale ranging from *never* (1) to *very often* (4), with the overall scale score ranging from 2 to 8. In the current sample, Cronbach's alpha was 0.75.

### 9.9. Medical Mistrust

The seven-item Medical Mistrust Index (MMI; LaVeist et al. 2009) was used to evaluate participants' mistrust of the medical system. Responses to the MMI were made on a 4-point Likert-type scale ranging from *strongly disagree* (1) to *strongly agree* (4). Scores range from 7 to 28. An example item includes, "Mistakes are common in health care organizations". The reliability coefficient for the MMI was 0.84.

## 10. Results

### 10.1. Descriptive Statistics

The means and standard deviation are displayed in Table 1. Of the 102 participants, 82 endorsed experiencing at least one of the listed ACEs. Of the various ACEs reported, the divorce or separation of parents, living with someone who was a problematic drinker or alcoholic or used street drugs, and having a household member who was depressed or mentally ill or attempted suicide were the most commonly endorsed experiences. On the other hand, 92.2% of the participants reported experiencing at least 8 of the BCEs, and 53.5% endorsed experiencing all 10. The most reported BCEs included having beliefs that provided comfort, opportunities to have a good time, good neighbors, at least one caregiver who provided a sense of safety, and at least one good friend.

**Table 1.** Means and standard deviations of study variables.

| Measures | *M* | *SD* |
|---|---|---|
| ACEs | 2.27 | 2.35 |
| BCEs | 9.30 | 0.91 |
| Anxiety Symptoms | 5.72 | 7.22 |
| Depression Symptoms | 3.81 | 4.43 |
| STS Symptoms | 23.03 | 5.35 |
| PTSD Symptoms | 0.52 | 1.00 |
| Negative Religious Coping | 6.91 | 5.14 |
| Negative Religious Interaction | 3.54 | 1.56 |
| Medical Mistrust | 18.45 | 3.82 |

For the anxiety, depression, and STS symptom scales, analyses were completed to understand the prevalence of moderate to severe scores among the participants. Twenty-eight

percent endorsed moderate to severe anxiety symptoms. Twelve percent endorsed moderate to severe depressive symptoms. Of the 23 participants who completed the measure of STS, 52% endorsed moderate to severe STS. Moreover, 26.5% reported experiencing at least one symptom of PTSD.

### 10.2. Correlation Analyses

Correlational analyses were run among all the scale variables. See Table 2 for bivariate correlations. The analyses determined that ACEs were significantly negatively associated with BCEs. ACEs were significantly positively associated with anxiety, depression, and PTSD. STS symptoms were significantly positively correlated with negative religious interaction, anxiety, depression, and PTSD. Anxiety symptoms were significantly positively associated with depressive symptoms. Depressive symptoms were significantly positively correlated with PTSD. Finally, negative religious coping was significantly positively correlated with negative religious interaction.

**Table 2.** Bivariate correlations.

| | 1 | 2 | 3 | 4 | 5 | 6 | 7 |
|---|---|---|---|---|---|---|---|
| 1. ACEs | - | | | | | | |
| 2. BCEs | −0.347 ** | - | | | | | |
| 3. STS | 0.104 | −0.165 | - | | | | |
| 4. Anxiety | 0.200 * | −0.096 | 0.553 ** | - | | | |
| 5. Depression | 0.236 * | −0.064 | 0.484 * | 0.692 ** | - | | |
| 6. PTSD | 0.369 ** | −0.018 | 0.480 * | 0.188 | 0.297 ** | - | |
| 7. Negative religious coping | 0.106 | −0.148 | 0.070 | 0.149 | 0.195 | 0.065 | - |
| 8. Negative religious interaction | −0.033 | −0.150 | 0.630 ** | 0.031 | 0.076 | 0.072 | 0.283 ** |

Note. ** = $p < 0.01$, * = $p < 0.05$.

### 10.3. Moderating Effect of Negative Religious Coping and Medical Mistrust

A regression model was run to assess the moderating role of negative religious coping on the relationship between ACEs and PTSD. The results revealed a positive and significant moderating impact of negative religious coping on the relationship between ACEs and PTSD (b = 0.020, t = 2.091, p = 0.039). The moderation analysis summary is presented in Table 3. Moreover, Table 4 shows the means and standard deviations for the specific items on the Negative Religious Coping scale. Based on the mean scores, participants reported wondering whether God had abandoned them, feeling punished by God for their lack of devotion, deciding the devil made a critical life event happen, and wondering what they did for God to punish them most often.

The results of a simple slope analysis conducted to better understand the nature of the moderating effects are shown in Figure 1. As displayed in Figure 1, the line is much steeper for high negative religious coping, demonstrating that at a high level of negative religious coping, the impact of ACEs on PTSD is much stronger in comparison to low negative religious coping. As shown in Figure 1, as the level of negative religious coping increases, the strength of the relationship between ACEs and PTSD increases as well.

**Table 3.** Results from a regression analysis examining the moderation of the effect of ACEs (X) on PTSD ($Y_1$) through negative religious coping (M).

| | | **Coeff.** | *SE* | *t* | *p* |
|---|---|---|---|---|---|
| Intercept | $i_1$ | 0.502 | 0.093 | 5.386 | 0.000 |
| ACEs (X) | $b_1$ | 0.147 | 0.042 | 3.533 | 0.001 |
| Negative religious coping (M) | $b_2$ | 0.011 | 0.019 | 0.609 | 0.544 |
| Moderation effect (X x M) | $b_3$ | 0.020 | 0.009 | 2.091 | 0.039 |
| | | $R^2 = 0.174$, $MSE = 0.844$ | | | |
| | | $F(3, 94) = 6.681$, $p < 0.001$ | | | |

Note. $N = 98$. *SE* = standard error. ACEs (X) and negative religious coping (M) were mean-centered before analysis.

**Table 4.** Means and standard deviations of negative religious coping items.

| | *M* | *SD* |
|---|---|---|
| Wondered whether God abandoned me. | 1.32 | 1.08 |
| Felt punished by God for my lack of devotion. | 1.26 | 1.02 |
| Wondered what I did for God to punish me. | 1.05 | 1.03 |
| Questioned God's love for me. | 0.79 | 1.04 |
| Wondered whether my church had abandoned me. | 0.80 | 0.94 |
| Decided the devil made this happen. | 1.13 | 0.98 |
| Questioned the power of God. | 0.55 | 0.95 |

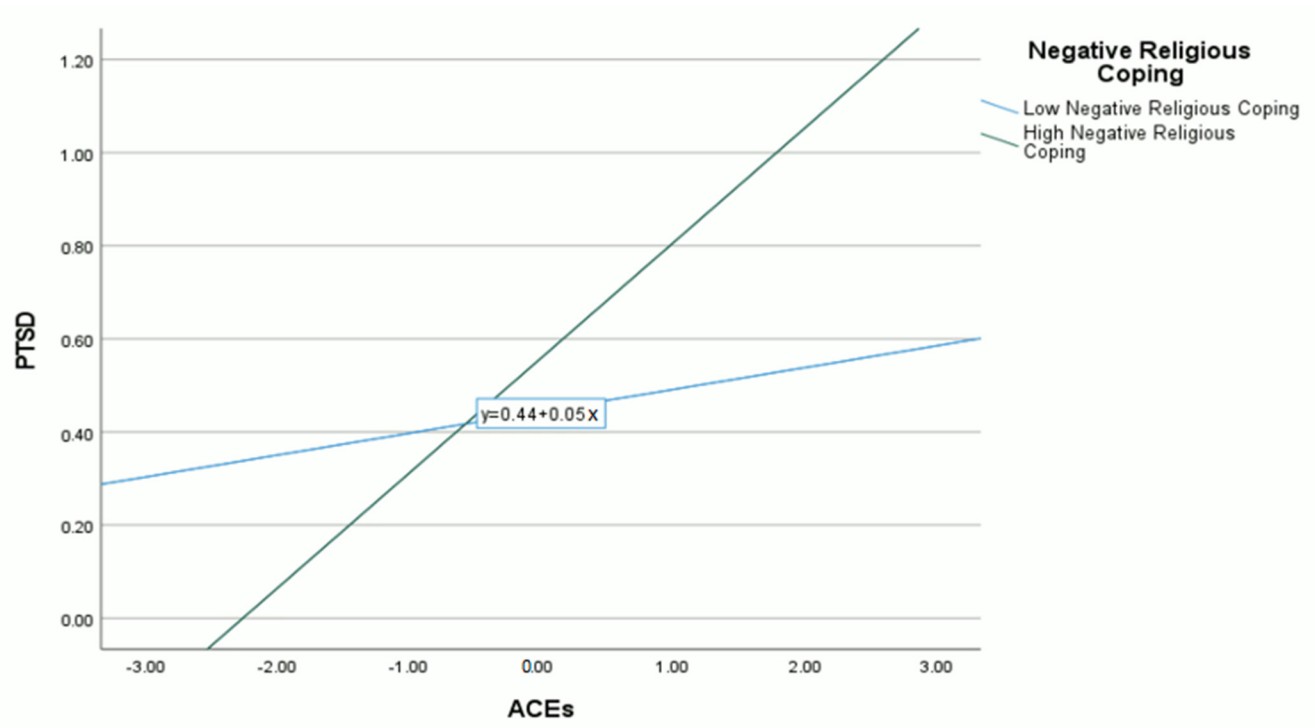

**Figure 1.** Simple slope from a regression analysis examining the moderation of the effect of ACEs (X) on PTSD ($Y_1$) through negative religious coping (M).

A second regression model was run to assess the moderating role of medical mistrust on the relationship between ACEs and PTSD. The results revealed a positive and significant moderating impact of medical mistrust on the relationship between ACEs and PTSD (b = 0.020, t = 2.471, *p* = 0.015). The moderation analysis summary is presented in Table 5.

Figure 2 shows the results of a simple slope analysis, which was conducted to better understand the nature of the moderating effects. Again, it is evident that the line is much steeper for high medical mistrust, demonstrating that at a high level of medical mistrust, the impact of ACEs on PTSD is much stronger in comparison to low medical mistrust. As shown in Figure 2, as the level of medical mistrust rises, the strength of the relationship between ACEs and PTSD rises as well.

**Table 5.** Results from a regression analysis examining the moderation of the effect of ACEs (X) on PTSD ($Y_1$) through medical mistrust (M).

| | | Coeff. | SE | t | p |
|---|---|---|---|---|---|
| Intercept | $i_1$ | 0.484 | 0.093 | 5.208 | 0.000 |
| ACEs (X) | $b_1$ | 0.145 | 0.042 | 3.424 | 0.001 |
| Medical mistrust (M) | $b_2$ | −0.023 | 0.026 | −0.893 | 0.374 |
| Moderation effect (X x M) | $b_3$ | 0.020 | 0.009 | 2.471 | 0.015 |
| | | $R^2 = 0.190$, $MSE = 0.830$ | | | |
| | | $F(3, 94) = 7.362$, $p < 0.001$ | | | |

Note. $N = 98$. SE = standard error. ACEs (X) and medical mistrust (M) were mean-centered before analysis.

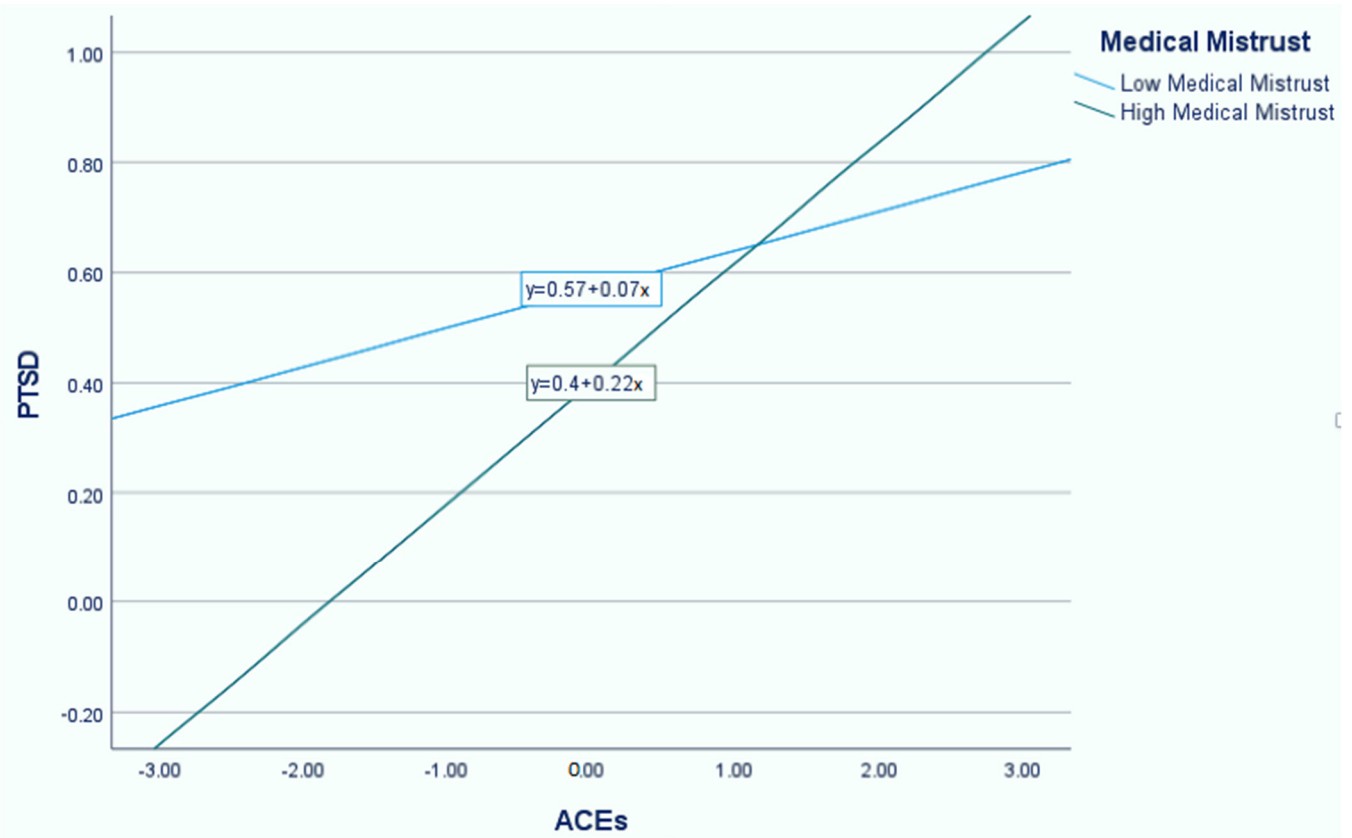

**Figure 2.** Simple slope from a regression analysis examining the moderation of the effect of ACEs (X) on PTSD ($Y_1$) through medical mistrust (M).

### 10.4. Effect of Negative Religious Interaction on STS

A simple regression analysis was used to examine the impact of negative religious interactions on STS in the original cohort of leaders. The results indicated that negative religious interactions have a significant and positive impact on STS (b = 2.034. 3.720, $p = 0.001$). Furthermore, the $R^2 = 0.397$, indicating that the model accounts for 39.7% of the variance in STS. The results are presented in Table 6.

**Table 6.** Results from a regression analysis examining the impact of negative religious interaction on STS.

| | Coeff. | t | p |
|---|---|---|---|
| Negative Religious Interaction | 2.034 | 3.720 | 0.001 |
| | $R = 0.630$, $R^2 = 0.397$ | | |
| | $F(1, 21) = 13.835$ | | |

## 11. Discussion

The present study aimed to explore the prevalence of and relationships among ACEs, BCEs, negative religious coping and church interactions, attitudes toward the medical system, and various negative mental health outcomes in African American religious leaders and church congregants. Given the likelihood of African American church congregants to seek out support from their religious leaders as well as the probability that these church leaders have experienced their own trauma, a variety of negative mental health symptoms have been documented among both African American religious leaders and their church congregants. The literature has documented the prevalence of STS in religious leaders (Hendron et al. 2011). Furthermore, Baum (2014) found that when professionals faced double exposure to the shared traumatic reality of the times, their mental health presentation consisted of a combination of primary traumatization, secondary traumatization, and general symptomology of distress. Research also suggests that African Americans exhibit depression with increased severity and persistence and PTSD at a higher rate than those of different races (Pérez Benítez et al. 2014; Roberts et al. 2011). Based on the findings of the literature, the current study expected that moderate to high levels of ACEs, STS, PTSD, depression, and anxiety would be present among African American leaders and their church congregants.

Descriptive statistics revealed that 80% of the sample experienced at least one type of adverse childhood experience, and 30.9% experienced three or more, compared to 61.5% and 24.6%, respectively, of a national sample (CDC). With a mean ACEs score of 2.27, the participants of this study averaged higher ACEs than a nationally representative sample of Black persons in the U.S. (1.69) (Merrick et al. 2018, p. 1042). The most common ACEs included the divorce or separation of parents, living with someone who was a problematic drinker or alcoholic or used street drugs, and having a household member who was depressed or mentally ill or attempted suicide.

Moreover, over half of the African American religious leaders endorsed experiencing moderate to severe levels of STS. The study further discovered that 28% of the African American pastors and congregants endorsed moderate to severe anxiety symptoms, 12% endorsed moderate to severe depressive symptoms, and 26.5% endorsed experiencing at least one symptom of PTSD on a checklist of five symptoms. Bivariate correlations specifically indicated that ACEs are positively correlated with anxiety symptoms among this population. Moreover, STS was positively correlated with anxiety, depression, and PTSD in the initial sample of religious leaders, which indicated that participants were experiencing distress related to being both a shepherd of a congregation as well as someone with his or her own challenging life experiences. The relationship of STS in the religious leaders with these negative mental health outcomes further suggests that the symptoms of depression and anxiety within this population could be trauma-based, whether it be based on personal traumatic experiences or secondary trauma. As previously mentioned, there is a lack of research on clergy and STS, but the findings of this study corroborate qualitative studies that show more attention needs to be given to the experience of clergy with STS (Francis et al. 2004; Hendron et al. 2011).

The literature has documented African American clergy members as key personnel for their congregants in times of crisis. Specifically, Black women shared in a semi-structured interview that clergy and other members of the church helped them deal with a crisis (Davis and Johnson 2020; St. Vil et al. 2016). Likewise, the results of the present study indicated that interaction with those who have experienced trauma and are currently experiencing negative mental health symptomology is a significant aspect of the African American church experience. Trauma exposure, ACEs, depression, anxiety, STS, and PTSD have associations with loss of faith in organized religion, interpersonal conflict, substance abuse, health issues, suicidal ideation, and overall general distress (Williams and Mohammed 2009; Hendron et al. 2011; McCormick et al. 2017). Thus, ACEs, depression, anxiety, STS, and PTSD are all prevalent within the African American church and should not be ignored.

Despite the troubling prevalence of ACEs and STS, it is important to recognize the large number of BCEs reported as well. Most of the participants (92.2%) endorsed eight or more BCEs. The most reported BCEs included having beliefs that provided comfort, opportunities to have a good time, good neighbors, at least one caregiver who provided a sense of safety, and at least one good friend. Conversely, the most commonly unendorsed BCEs included liking or feeling comfortable with oneself, having an adult other than a parent/caregiver to provide advice and support, and liking school.

It is noteworthy that despite having higher ACEs scores than average and relatively high STS scores, the average participant did not meet the cutoff scores for anxiety and depression. The mean score on the GAD-7 for anxiety was 5.72, which places this group of participants at the low end of mild anxiety, and the average depression score on the PHQ-9 was 3.81, which is at the high end of minimal depression. As shown in a growing number of studies, the abnormally high rates of BCEs for this group may have served as a protective factor against at least two of the most common mental health diagnoses (Bethell et al. 2019; Narayan et al. 2021). Most BCEs are relational in nature and can be found within growth-fostering communities. The church is potentially one such community as it often provides children with experiences of having at least one good friend, obtaining beliefs that provide comfort, an adult who is not in the home who can provide support or advice, and opportunities to have a good time. These are all factors found to be benevolent childhood experiences.

Throughout history, the African American church has undertaken roles that mirror BCEs. It has provided a place of refuge, social support, advocacy, validation, and tangible resources such as food, shelter, financial assistance, and further direction. Given the protective nature of BCEs documented throughout the literature (Crandall et al. 2019; Doom et al. 2021; Narayan et al. 2021), it is encouraging to discover the high prevalence of BCEs documented throughout this sample. Moreover, the church can continue to foster a community that provides advice, support, and direction for its congregants across all age groups to continue this trend.

Many unique stressors accompany leading a church. The literature has documented the emotional and mental challenges faced by clergy who experienced high demands from their church congregations (Doolittle 2010; Jacobson et al. 2013; Roberts et al. 2011). The current study discovered that the unfair expectations that the African American religious leaders felt their congregants put on them predicted STS symptoms. Specifically, religious leaders who felt that the people in the congregation made too many demands on them and/or were critical of them and the things they did reported increased STS. Vocational STS has been associated with departing from the ministry, emotional exhaustion, and relational difficulties (Spencer et al. 2012; Hendron et al. 2011). Thus, the demands and criticisms communicated to religious leaders may threaten not only their well-being but their longevity in the ministry as well.

Throughout the literature, spirituality has been thought of as beneficial to one's overall health and well-being, often focusing on the protective aspects of religious coping and connection to God. However, research has also discovered a relationship between negative religious coping and higher levels of distress (Park et al. 2018). In a study of African Americans who had experienced trauma, religious coping did not reduce PTSD and depressive symptoms, leading researchers to hypothesize potential harmful uses of religious coping (Bryant-Davis et al. 2011). The researchers proposed that the participants may have been utilizing negative religious coping (Bryant-Davis et al. 2011). While positive religious coping emphasizes the benevolence of God, negative religious coping focuses primarily on a condemning God (Pargament et al. 2011). In the current study, negative religious coping in times of challenges was characterized by feeling punished by God, a sense of abandonment from God and the church, blaming hardship on the devil, and questioning the power and love of God.

The present study discovered that negative religious coping moderated the relationship between ACEs and PTSD (Bryant-Davis et al. 2011; Bryant-Davis et al. 2015). African

American church leaders and congregants who reported using negative religious coping in times of hardship were at greater risk of PTSD following ACEs, which provides further support for the findings of Bryant-Davis et al. (2011). Previous research has suggested, and the current study substantiates, that individuals might consider re-evaluating their understanding of God following challenging life events because their implicit and/or explicit view of God may exacerbate vulnerabilities to mental illness. Notably, the significant moderation effect applied to individuals demonstrating high levels of negative religious coping; for those demonstrating low and average levels of negative religious coping, such coping did not moderate the relationship between ACEs and PTSD. Therefore, negative religious coping is only a risk factor that exacerbates the relationship between ACEs and PTSD for those who frequently engage in it.

These findings suggest that religion does not inherently lead to positive mental health outcomes. The way one approaches their religiosity and spirituality, particularly when responding to stressful life events, is an important consideration, as not all forms of religious coping are beneficial. Traumatic events can leave individuals struggling to make meaning of their situation and the world. Religion is often a context that people lean upon to make sense, find meaning, and recover from challenging circumstances throughout their lifespan (Bryant-Davis et al. 2011; De Luna and Wang 2021). Though it does not appear to be harmful for individuals to question God's benevolence every so often, frequent feelings of abandonment and punishment from God and the church, as well as other forms of negative religious coping, appear to result in poorer outcomes. More specifically, in response to dealing with a critical life event, participants in the current study most often reported wondering whether God had abandoned them, feeling punished by God for their lack of devotion, deciding the devil made it happen, and wondering what they did for God to punish them. African American church leaders may keep in mind that their congregations may benefit from support in coping and making sense of stressful life events, particularly with those who have been exposed to trauma.

Given the history of racism and barriers to healthcare, mistrust of the medical system and hesitancy to seek mental health support is often present among African Americans. Research indicates that African Americans experience lower rates of recovery from PTSD, resulting from several disparities. However, even after controlling for several predisposing factors, research has indicated that African American patients are less likely to have had an appointment with a mental health provider (Fiscella et al. 2002; Hays 2015). Moreover, a longitudinal study of African Americans with anxiety revealed that those with PTSD were less likely to seek treatment (Pérez Benítez et al. 2014). Finally, research has suggested that African Americans experience increased feelings of mental health stigma, negative attitudes toward mental health treatment, and fears of discrimination in a mental health setting (Williams 2008; Williams et al. 2014). The reasonable and adaptive reactions of many African Americans to healthcare lead to greater dependence on the Black church for mental health support (Adofoli and Ullman 2014; Hankerson and Weissman 2012; Hays 2015).

Though previous research has postulated medical mistrust, mental health stigma, limited interactions with mental health providers, and increased mental health symptomology among African Americans, the specific relationship among these variables has not yet been determined (Fiscella et al. 2002; Hays 2015; Pérez Benítez et al. 2014). Based on the results of this study, it appears that the mental health of African American churchgoers with varying numbers of ACEs is impacted by medical mistrust. Though low and average amounts of distrust of healthcare organizations have an insignificant impact on PTSD, high levels of medical mistrust are a risk factor moderating the relationship between ACEs and PTSD. It may be beneficial for future research to further explore the relationship between medical mistrust and mental health symptoms to understand the specific avenues for this finding. However, it can be postulated that those with high levels of medical mistrust may be underutilizing health care services and, therefore, not getting the mental and physical care needed.

The moderating relationships of both negative religious coping and medical mistrust on the relationship between ACEs and PTSD have important implications for the health and well-being of African American leaders and churchgoers. The findings of the current study offer two empirically supported interventions that can be used in reducing PTSD symptoms in this population. Mental health clinicians and church leaders can focus efforts on diminishing negative religious coping and medical mistrust in African American Christians. In efforts to promote culturally competent treatment when working with African Americans, therapists can employ interventions that aim to reduce negative religious coping. Specifically, therapists can integrate spiritual and religious issues related to clients' feeling punished by God and abandoned by God and the church as well as blaming hardship on the devil and questioning the power and love of God. Additionally, both clinicians and trusted church personnel can provide psychoeducation designed to correct misconceptions regarding the medical system. Churches may consider partnering with healthcare institutions to build relationships and overcome some of the barriers. However, the results ultimately shed light on the imperativeness of cultivating a medical system that warrants the trust of African Americans.

*Limitations and Directions for Future Research*

There are several limitations to the present study. First, the data were collected from a sample of African American churches in the state of Texas, and most of the sample (69.7%) identified as female. Therefore, the findings may be impacted by the cultural context of the specific region and gender, affecting the universality of the results. Furthermore, the study may have been impacted by self-selection and self-report biases. It is possible that only those interested in the topic chose to participate. Potential respondents with higher levels of symptomatology may not have wanted to participate due to the nature of the survey questions or current life stressors. Moreover, this study solely used self-report measures, which are subject to various aspects of bias by the respondents. Though the measures utilized in this study have good reliability and validity, most of the surveys were shortened to a few items or subscales to reduce the time commitment for the participants. Finally, the symptom severity of respondents was very diverse, which initially resulted in skewed data. Therefore, outliers had to be deleted for high depression and PTSD scores as well as low BCEs scores.

As this study confirmed the prevalence of STS, anxiety, depression, and PTSD among African American religious leaders and church congregants, future research should continue to explore the best avenues of support for this population. Possible constructs to be explored in relation to trauma and mental health in the African American church include positive religious coping, spiritual experiences, experiences of discrimination, and education surrounding trauma and mental health. Trauma exposure may also be expanded in future research to include experiences after the age of 18 as well. Future research might also investigate the variables that make one more at risk for high levels of negative religious coping and medical mistrust. Additionally, future research can also explore variables (especially those that are religiously oriented) that may contribute to post-traumatic growth, such as self-compassion (Yuhan et al. 2021) and different forms of prayer (Lowe et al. 2022).

## 12. Conclusions

In conclusion, this study expands on the literature by further substantiating the existence of STS, depression, anxiety, and PTSD among African American religious leaders and church congregants. Additionally, the study contributes to the literature by indicating the relationship between STS and negative religious interactions. Finally, it adds to the current literature by addressing how negative religious coping and medical mistrust moderate the relationship between ACEs and PTSD in African American religious leaders and congregants. Given these findings, future researchers can continue to uncover mechanisms to support the well-being of African American religious leaders and their congregants with the hope to reduce barriers to mental health care in this population.

**Author Contributions:** Conceptualization, L.R., D.C.W., L.D. and E.M.B.; methodology, L.R., D.C.W., L.D., E.H., B.A.L. and E.M.B.; writing—original draft preparation, L.R.; writing—review and editing, D.C.W., L.D., E.H., B.A.L. and E.M.B.; supervision, D.C.W., L.D. and E.M.B. All authors have read and agreed to the published version of the manuscript.

**Funding:** This research was funded by the Lilly Endowment, Inc., grant number 2020 0849.

**Institutional Review Board Statement:** The study was approved by the Institutional Review Board of Biola University (F21-011_JB; 13 September 2021).

**Informed Consent Statement:** Informed consent was obtained from all subjects involved in the study.

**Data Availability Statement:** Not applicable.

**Conflicts of Interest:** The authors declare no conflict of interest. The funders had no role in the design of the study; in the collection, analyses, or interpretation of data; in the writing of the manuscript; or in the decision to publish the results.

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
