# Peer review of "Secondary Traumatic Stress, Religious Coping, and Medical Mistrust among African American Clergy and Religious Leaders"

_religions, doi:10.3390/rel14060793_

Round 1

Reviewer 1 Report

This is a strong contribution to understanding a situation that has been important in discussing medical ethics, psychology and the African American religious environment. It became even more important during the pandemic, because African-American communities were hit especially hard by Covid (at least in the urban USA). How clergy become mediators between the medical-scientific establishment and the people of their congregations has been part of that picture. Here, the authors focus explicitly on the personal challenges, stresses and resilience of those local religious leaders. "Secondary traumatic stress" is the term that highlights the particular pressures clergy face, especially when they are among the very few authoritative and trustworthy sources for help - or at least perceived as that by their needy congregations. The authors provided a clear and fair-minded introduction to this landscape, where mistrust of medical expertise and institutions has a long history.  Part of this story is its complicated relations to the faith traditions of most African-Americans. (Note: only Christian religious leaders were studied in this essay.)  I found the authors' treatment of this situation really readable and excellent. Their own contribution is to try to untangle the complicated relationships between stress from dealing continuously with others' traumas (as well as their own), professional roles that put them always in the spotlight, and their religious resources. Some of the latter are unhelpful, and seem to add to stress. The authors recognize that African American clergy and lay religious leaders have been neglected in studies of stress and traumas' impact on counselors and therapists. They are trying to correct for this omission. 

After reading this fine study, I recalled hearing that as the pandemic waned, 25% of all clergy planned to leave ministry and pursue other careers. The stresses had become overwhelming for them. I do not know if this statement is accurate, nor if it applies less to African-American clergy. But in any case, this essay is especially timely.

Author Response

Thank you for your very helpful and encouraging feedback on our paper.  How you read this paper was exactly what we were hoping to achieve (i.e., to address gaps in the literature on the experiences of African-American clergy as they navigate stressors from dealing continuously with others' traumas (as well as their own), professional roles that put them always in the spotlight, etc...).

Reviewer 2 Report

I find the article well-written and well-argued 

My only concern is that throughout the article the author/s makes general references to sources without indicating the specific pages in the sources where readers can verify the data and locate it for further reading. I feel that instead of just stacking up sources, the article will be better with specific maps to sources used. 

Author Response

Thank you for your encouragement.  Agreed.  While this academic paper does indeed point the reader to a plethora of resources (particularly in our discussion section), because of the nature and genre of this form of literature, we were not able to include a map to practical resources.  As our research and resourcing initiative continues (we are only in year 2 of a 5-year initiative), we are planning to put together the kind of resource map that you suggested here.

Reviewer 3 Report

The article aims to focus on secondary traumatic stress within African American clergy in Texas, USA. In their role as religious leaders, they provide a means of linkage between the local communities and various social services. The subject of the article is novel and rarely studied in empirical research. It is set in the broader framework of STS in religious leaders in general and then in the specific context of African American communities both within religious leaders and congregants. The literary review encompasses an extensive bibliography and it is clear and well-organized. The empirical research focuses on the role of religious leaders within the Afro-American churches of Texas. The research is well articulated; it presents the tools and methods and, in a brief but precise formulation, the scales used. Results are accurate and the discussion is careful and critical and compares itself with the current literature on the topic. The authors are openly aware of the limitations of the research, such as recruiting participants from a network of churches in Texas and collecting data by self-selection and self-report. However, the report, as it is, is well thought out. That said, in order to widen the scope and give further value to the research, the authors could have explored the impact of different religious congregations. One more noteworthy limitation of the research is the failure to present the differences in outcome between male and female participants, which in the literature review is suggested as having an impact on the variables considered. Overall, the article, in its current form, is engaging and believed to provide a valuable contribution to the field, hence its publication is recommended.

Author Response

Thank you for your helpful and encouraging feedback as well.  We agree with both the strengths as well as the limitations noted and have updated our manuscript to highlight the particular limitations that you mentioned in more detail and suggesting that future research address these limitations.